# Assessment of the Severity and the Remission Criteria in Eosinophilic Esophagitis

**DOI:** 10.3390/biomedicines11123204

**Published:** 2023-12-01

**Authors:** Ksenia Maslenkina, Liudmila Mikhaleva, Alexander Mikhalev, Valeria Kaibysheva, Dmitri Atiakshin, Eugeny Motilev, Igor Buchwalow, Markus Tiemann

**Affiliations:** 1A.P. Avtsyn Research Institute of Human Morphology, Petrovsky National Research Center of Surgery, 119991 Moscow, Russia; ksusha-voi@yandex.ru (K.M.); mikhalevalm@yandex.ru (L.M.); evgeny_2008_@mail.ru (E.M.); 2Laboratory of Surgical Gastroenterology and Endoscopy, Pirogov Russian National Research University, 117997 Moscow, Russia; rsmu1985@gmail.com (A.M.); valeriakai@mail.ru (V.K.); 3Research and Educational Resource Center for Immunophenotyping, Digital Spatial Profiling and Ultrastructural Analysis Innovative Technologies, RUDN University, 6 Miklukho-Maklaya St., 117198 Moscow, Russia; buchwalow@pathologie-hh.de; 4Institute for Hematopathology, Fangdieckstr. 75a, 22547 Hamburg, Germany; mtiemann@hp-hamburg.de

**Keywords:** eosinophilic esophagitis, activity, severity, remission, EREFS, EoEHSS, I-SEE

## Abstract

Eosinophilic esophagitis (EoE) is an immune-mediated disease that manifests with dysphagia and is characterized by the predominantly eosinophilic infiltration of the esophageal mucosa. Several instruments have been developed to assess the symptoms of EoE: the Daily Symptom Questionnaire (DSQ), EoE Activity Index (EEsAI), Pediatric EoE Symptom Severity (PEESSv2), etc. The use of the EREFS is a gold standard for endoscopic diagnosis. The EoE histologic scoring system (EoEHSS) was elaborated for the assessment of histological features in EoE. However, the remission criteria are not clearly defined and vary greatly in different studies. Gastroenterologists establish the severity of EoE mainly based on endoscopic findings. At the same time, EoE requires a multidisciplinary approach. The recently developed Index of Severity of Eosinophilic Esophagitis (I-SEE) that is built on symptoms, endoscopic findings, and histological features is promising.

## 1. Introduction

Eosinophilic esophagitis (EoE) is an immune-mediated disease of the esophagus that manifests with dysphagia and is characterized by the predominantly eosinophilic infiltration of the esophageal mucosa (>15 eosinophils in high power field, eos/hpf) [1,2,3,4,5]. EoE is a chronic disease arising in predisposed individuals that often harbor one or several atopic conditions: bronchial asthma, atopic rhinitis, atopic dermatitis, eczema, etc. [6,7,8]. EoE may present with seasonal exacerbations due to aeroallergen exposure [9,10].

Almost half of the patients who were diagnosed in childhood do not have any symptoms 8 years after their diagnosis [11,12], though 2/3 of them did not receive drug therapy. At the same time, there is a risk of strictures increasing over time in patients who manifest in adulthood [13]. The more time that passes without treatment, the more the risk of strictures rises. Strictures were identified in 17% of patients that were diagnosed during 2 years from the first manifestation, in 31% of patients who were diagnosed 2–5 years after the first manifestation, in 38% patients with a delay in diagnosis for 8–11 years, in 64% of patients with a delay in diagnosis for 14–17 years, and in 71% of patients with a delay in diagnosis for more than 20 years. In another study, diagnostic delay for every year was associated with an increase in stricture presence for 9% [14]. Three phenotypes of the EoE were identified: inflammatory, mixed, and stenofibrotic [15]. The inflammatory phenotype was more common in young ages (13 vs. 29 vs. 39 years, respectively; *p* < 0.001), and dysphagia and food impaction were rare in this phenotype. Frequent symptoms included abdominal pain, vomiting, and failure to thrive. Individuals with the inflammatory phenotype more often reported atopy. The rate of stenosis increased with age (for every 10 years, the odds ratio of stenosis was 2.1 (95% CI, 1.7–2.7)). The risk of stenosis increased by 5% with every year of symptoms prior to the diagnosis. Therefore, EoE shows the progression from the inflammatory to stenofibrotic phenotype.

The treatment of EoE is focused on the induction of the sustained remission that implies not only the absence of symptoms, but the absence of histological features of the disease [16]. The management of EoE is based on diet and drug therapy that includes proton pump inhibitors (PPIs), topical steroids, and biological therapy (with antibodies to IL-4, IL-5, IL-13, etc.) [17,18,19,20,21]. Esophageal strictures can be successfully treated with dilation.

Diet therapy includes an elemental diet and empiric food elimination. An elemental diet represents drinking a special elemental formula consisting of amino acids, fats, sugars, vitamins, and nutrients. In a systematic review of six observational studies, an elemental diet was associated with histological remission (<15 eos/hpf) in 93.6% of patients compared to 13.3% in the placebo group (RR, 0.07; 95% CI, 0.05–0.12) [22]. The elemental diet intervention has the highest response rate compared to the other dietary interventions. However, this approach is costly, difficult to adhere to because of taste, and may be associated with the development of IgE-mediated food allergies during food reintroduction [23]. Empiric food elimination diets are based on the elimination of the most common foods causing the allergy-mediated inflammation of the esophagus in EoE. This approach includes a six-food elimination diet (SFED, with the elimination of milk, wheat, eggs, soy, peanuts/tree nuts, and fish/shellfish), a four-food elimination diet (restricting milk, wheat, legumes, and eggs), and other less restrictive diet regimes. SFED is the most studied approach, with a histological response in 67.9% of patients (<15 eos/hpf) compared to 13.3% in the placebo comparison group (RR 0.38; 95% CI, 0.32–0.43) in the systematic review [22]. Dietary adherence and the need for repeated endoscopies during food reintroduction are the main difficulties of this approach.

PPIs represent the first drug of choice in EoE because they are inexpensive, easy to administer, and have minimal side effects. High-dose PPI treatment led to the complete resolution of symptoms in 68% of patients, with EoE and histological remission (<15 eos/hpf) in 49% of patients [24]. In a systematic review of 23 observational studies with 1051 EoE patients, PPI therapy gave a histological response (<15 eos/hpf) in 41.7% of patients compared to 13.3% in the placebo group (RR, 0.66; 95% CI, 0.61–0.72) [22]. The mechanisms of action of PPIs in EoE are independent of gastric acid suppression and include the inhibition of immune cell function and the reduction in allergic Th2 inflammation, reduction in epithelial cell inflammatory cytokine expression, and the partial restoration of the integrity of the epithelial barrier [25,26,27,28].

Topical corticosteroids (TCS) are also widely used for the treatment of EoE. A systematic review of eight double-blind placebo-controlled clinical trials of TCS treatment that included 437 patients showed that TCS were associated with histologic remission in 64.9% of patients (<15 eos/hpf), compared to 13.3% in patients treated with the placebo (RR 0.39, 95% CI, 0.85–1.19) [22]. Budesonide orally disintegrating tablets (BOT) demonstrated the induction of histological and clinical remission in 90.1% and 75% of patients at 6 weeks, respectively [29]. Corticosteroids modulate the NF-kB inflammatory cascade and directly decrease the expression of eotaxin-3, thereby reducing the recruitment of eosinophils in the esophagus [30]. TCS are well tolerated. Side effects from TCS include oral or esophageal candidiasis [31] and exclusively rare adrenal insufficiency [32].

Severe esophageal structures and a narrow caliber esophagus need dilation. A meta-analysis of 27 studies with 845 EoE patients who underwent a total of 1820 esophageal dilations demonstrated that dilation improved dysphagia in 95% of patients following the procedure (95% CI = 90–98%) [33]. The complications included perforation, which occurred in 0.38% (95% CI: 0.18–0.85), hemorrhage, which occurred in 0.05% (95% CI: 0–0.3%), and hospitalization in 0.67% (95% CI: 0.3–1.1%). Another systematic review and meta-analysis of 37 studies including 977 patients with EoE that underwent 2034 dilations showed that dilation was associated with a perforation rate of 0.033% (95% CI, 0–0.226%), a bleeding rate of 0.028% (95% CI, 0–0.217%), and a hospitalization rate of 0.689% (95% CI 0–1.42%) [34]. At the same time, dilation does not have an impact of esophageal inflammation, but rather, it resolves dysphagia symptoms by widening the esophageal diameter. That is why ongoing esophageal inflammation and remodeling requires repeated dilations.

So, there are different therapeutic options to treat EoE. Nevertheless, the terms “clinical remission”, “endoscopic remission”, and “histological remission” define different entities and do not always coincide. For example, clinical symptoms may persist in patients with histological remission [35,36,37]. Currently, the medical community lacks a uniform definition and uniform criteria of the remission in EoE [18,38,39]. Different endpoints are used in different studies to detect the clinical efficacy of therapy. Different definitions of clinical, endoscopic, and histological remissions are reviewed in this article.

## 2. Clinical Symptom Assessment in EoE

The clinical symptoms of EoE vary in patients of different age groups [15,40,41,42]. In little children, they include feeding problems, food refusal, nausea, vomiting, and failure to thrive; older children complain of abdominal pain, vomiting/regurgitation, and heartburn and may present with weight loss. In children, pain may also lead to sleeping problems [43]. Dysphagia and food impaction prevail in teenagers and adults. Indeed, in adults, EoE is the most frequent reason for dysphagia and food impaction [44]. Food impaction requiring endoscopy is observed in 33–53% of adults with EoE [45].

It should be mentioned that EoE implies changes in eating behaviors that involves the use of coping mechanisms to avoid symptoms. Patients eat more slowly, chew excessively, take smaller pieces, imbibe fluids with meals, and avoid hard-textured foods [18]. The use of these coping mechanisms hampers the assessment of symptoms [39].

Several patient-reported outcomes were developed to evaluate the severity of symptoms of EoE and the response to the therapy [38]: Daily Symptom Questionnaire (DSQ) [32,46,47,48,49], EoE Activity Index (EEsAI) [50,51], Pediatric EoE Symptom Severity (PEESSv2) [52], etc. Dellon ES et al. [46] developed DSQ, which includes three questions for the identification of dysphagia: (1) did you eat solid food this day, (2) has food gone down slowly or been stuck, and (3) did you have to do anything to make the food go down or to get relief? The DSQ results showed a correlation with the number of dysphagia days (R = 0.96; *p* < 0.001) and the Straumann Dysphagia Instrument (SDI, R = 0.77; *p* < 0.001). The DSQ was successfully validated [47]. The treatment of the oral suspension of budesonide (BOS) resulted in a substantial decrease in DSQ (from 29.3 to 15.0 in the BOS group vs. from 29.0 to 21.5 in the placebo group, *p* = 0.0096) [48]. A decrease in DSQ ≥ 30% was a criterion of therapy efficacy [32,49]. The dysphagia symptom diary (DSD) [53] is a modification of the DSQ, which contains a fourth question about the presence and severity of pain during dysphagia. A decrease in the symptom score was more prominent in the group receiving 360 mg RPC4046 than in the placebo group.

The EEsAI [50] includes an evaluation of the following parameters: duration, frequency, and severity of dysphagia, pain when swallowing, a visual dysphagia question concerning food consistency, and behavioral adaptations (avoidance, modification, and slow eating of various foods). The optimal time of symptom recall was 7 days. The values of the EEsAI were greater in active EoE in terms of endoscopic and histological features. EEsAI ≤ 20 allowed for the identification of patients with endoscopic remission with an accuracy of 65.1% and for the identification of patients with histological remission (<20 eos/mm^2^, which corresponds with <5 eos/hpf) with an accuracy of 62.1% [37]. The EEsAI was successfully validated and was used in different studies [29,51,54,55,56,57,58].

The Straumann Dysphagia Instrument (SDI) is a nonvalidated questionnaire that rates the frequency and intensity of dysphagia from 0 to 9 points [59]. The SDI was used to show the efficacy of budesonide in EoE (SDI decreased from 5.61 to 2.22, *p* < 0.0001). The SDI was used to evaluate clinical response in other studies [54,60,61]. A decrease in the SDI > 3 points was the criterion of symptomatic remission.

The Dysphagia Symptom Score (DSS) is a nonvalidated questionnaire, reflecting the frequency, intensity, duration of symptoms, and presence of lifestyle changes [62,63]. The Mayo Dysphagia Questionnaire (MDQ) is a validated instrument for the assessment of dysphagia in esophageal diseases [64,65]. It was not developed for EoE, and in clinical studies [36,66], the assessment of symptoms using the MDQ did not show a correlation with the induction of histological remission. The results of dysphagia assessment using a visual analogous scale (VAS) demonstrated a correlation with the Likert scale (R = 0.77; *p* < 0.0001) and MDQ (R = 0.46, *p* = 0.001) [67].

PEESSv2 was developed for children and their parents [52], and it includes an assessment of the following parameters: chest pain, heartburn, abdominal pain, dysphagia/food impaction, vomiting, nausea, regurgitation, and poor appetite. The PEESSv2 dysphagia domain showed a correlation with biological signs of activity in EoE [68], with the expression of eosinophil peroxidase in both distal and proximal biopsies (ρ = 0.23–0.37, *p* = 0.019–0.17), and with the expression of the tryptase and chymase of mast cells (ρ = 0.34, *p* = 0.036 and ρ = 0.32, *p* = 0.041, respectively). Moreover, there was an association between dysphagia and the expression of CPA3 (ρ = 0.36, *p* = 0.02). A transcriptome analysis revealed that dysphagia was associated with the expression of genes related to eosinophilia, chemokines, mast cells, neurosensory, cytokines, and inflammation.

Other patient-reported outcomes include the symptom scoring tool (SST) [69,70] and clinical symptom score (CSS) [71]. The SST [69] represents seven questions for the identification of heartburn/regurgitation, abdominal pain, nocturnal awakening, nausea/vomiting, anorexia/early satiety, dysphagia/odynophagia, and gastrointestinal hemorrhage. Dysphagia and anorexia/early satiety were associated with EoE (odds ratio 15 (95% CI, 6–68), *p* < 0.001). Symptoms of dysphagia and anorexia/early satiety are also correlated with the average epithelial score (r = 0.32, *p* < 0.05). Dysphagia is correlated with the maximum lamina propria score (r = 0.45, *p* < 0.05). In a randomized placebo-controlled study, the use of oral viscous budesonide was associated with a decrease in the severity of symptoms measured by SST (*p* = 0.0007) [70].

The CSS [71] evaluates the following symptoms in points from 0 to 3: (1) heartburn, (2) abdominal pain, (3) nocturnal awakening due to the symptoms, (4) nausea, regurgitation, or vomiting, (5) anorexia or early satiety, and (6) dysphagia, odynophagia, or food impaction. The clinical response was defined as a decrease in CSS ≥ 50%, and clinical remission was defined as 0 points of CSS. An assessment of compensatory mechanisms is an advantage of CSS, although the use of CSS did not allow for the differentiation between the oral budesonide suspension group and the placebo group.

To sum up, there are different instruments to evaluate the symptoms of EoE (Table 1), but not all of them are responsive to treatment and correlate with histological remission. While the criteria of the response to treatment vary in different studies, an intuitive definition of remission should be the absence of symptoms. Nevertheless, the absence of symptoms does not designate the absence of biological activity of the disease [37]; therefore, there is a need for a complex evaluation of symptoms, endoscopic features, and histological findings.

## 3. Evaluation of Endoscopic Features in EoE

Endoscopic findings in EoE include linear furrows (in 48% of patients), esophageal rings (in 44% of patients), pallor/decreased vascularity (in 41% of patients), white plagues (in 27% of patients), strictures (in 21% of patients), and narrow-caliber esophagus (in 9% of patients) [72]. Rings and strictures are more often identified in adults (57% and 25%, respectively) than in children (11% and 8%, respectively, *p* < 0.05). Plagues and pallor/decreased vascularity are more often seen in children (36% and 58%, respectively) than in adults (19% and 18%, respectively, *p* < 0.05).

The gold standard of endoscopic evaluation in EoE implies the use of the Endoscopic Reference Score (EREFS), which includes the assessment of the following features: edema, rings, exudates, furrows, and strictures (Table 2) [73]. The EREFS showed a moderate level of interobserver agreement (κ for different features varies from 0.50 to 0.58, the proportion of pairwise agreement for experts and non-experts was 81% and 74% for exudates, 90% and 77% for edema, and 98% and 90% for crepe paper esophagus. In another study [74], interobserver agreement was substantial for rings (κ 0.70), white exudates (κ 0.63), and crepe paper esophagus (κ 0.62), moderate for furrows (κ 0.49) and strictures (κ 0.54), and slight for edema (κ 0.12). The EREFS classification system identifies patients with EoE in an area under the receiver operator characteristic curve of 0.934 [75]. The threshold of a score of 2 allowed for EoE to be diagnosed with a sensitivity of 88%, specificity of 92%, positive predictive value of 84%, negative predictive value of 94%, and accuracy of 91%. The EREFS score decreased after treatment, and a more prominent reduction was observed in patients with histological response (<15 eos/hpf).

Rings, strictures, and crepe paper esophagus correspond to fibrotic lesions and exudates, while edema and furrows stand for inflammatory lesions [13,76]. Rodríguez-Sánchez J. et al. [63] showed that exudates (*p* = 0.03), furrows (*p* = 0.03), and inflammatory score (*p* < 0.001) predicted the histological activity of EoE, although only exudates correlated with histological response after treatment, while furrows and edema persisted in 50% and 70% patients with histological remission, respectively. Crepe paper esophagus, diffuse exudates, and severe rings correlated with a higher EREFS score.

Van Rhijn, 2016 [77] revealed that rings and furrows were more prominent in active EoE. Fibrotic scores (r = 0.28, *p* = 0.020), inflammatory scores (r = 0.27, *p* = 0.025), and total EREFS scores (r = 0.43, *p* < 0.001) correlated with the peak eosinophil count (PEC) in biopsies, although the reduction in PEC after treatment was not accompanied with the decrease in the EREFS score. Therefore, the predictive values of EREFS were not sufficient to predict the histological activity of the disease.

Chen J.W., 2016 [78] demonstrated that higher ring scores were associated with a lower distensibility plateau (rs = −0.46; *p* < 0.0001) and a higher risk of food impaction. Food impaction in anamnesis was observed in 20% patients with a ring score of 0, in 25% of patients with a ring score of 1, in 39% patients with a ring score of 2, and in 100% patients with a ring score of 3 (*p* = 0.002). The severity of exudates (r = 0.27; *p* = 0.02) and furrows (r = 0.49; *p* < 0.0001) correlated with the density of intraepithelial eosinophils in biopsies.

The detection of more than one EREFS feature identified children with EoE with 89.6% sensitivity and 87.9% [79]. The EREFS score correlated with PEC (*p* < 0.001). The EREFS score significantly reduced after treatment in children (2.4 vs. 0.7, *p* < 0.001) [79] and adults (3.88 vs. 2.01, *p* > 0.001) [75]. Therefore, EREFS is responsive to treatment. Moreover, the severity of endoscopic features is the main determinant of disease activity for gastroenterologists [80].

In most studies, the treatment results are described as a decrease in the EREFS score [32,48,49,53,54,57,58,60,75,79]. Safroneeva E. et al. [37] gave the following three definitions of endoscopic remission: endoscopic inflammatory remission, endoscopic fibrotic remission, and total endoscopic remission (inflammatory and fibrotic remission). Endoscopic inflammatory remission included the following: the absence of white exudates; furrows and edema may be present, but not in combination. Endoscopic fibrotic remission was defined as the absence of moderate and severe rings, and the absence of strictures. Total endoscopic remission included signs of both inflammatory and fibrotic remission. Greuter T. et al., 2019 [81] and Miehlke S. et al., 2022 [29] defined endoscopic remission as the absence of inflammatory features of EoE (exudates, furrows, and edema), while mild rings may be present. Thus, a uniform definition of endoscopic remission currently is not developed.

The functional lumen imaging probe (FLIP) evaluating distensibility plateau represents a technique to simultaneously measure the esophageal luminal cross-sectional area and distensive pressure during volume-controlled esophageal distension. Distensibility plateau was reduced in patients with EoE compared with asymptomatic controls [82]. A greater reduction in DP in patients with EoE was associated with the future risk for food impaction and/or requirement for therapeutic dilation [83]. That is why the distensibility plateau may reflect disease severity in EoE, and its improvement can be used as a measurement of clinical outcome.

## 4. Histological Criteria of the Remission in EoE

The diagnosis of EoE requires the presence of >15 eos/hpf [1,2,3]. However, the transitional eosinophilia of the esophageal mucosa may be attributed to caffeine exposure [84]. Currently, there is no uniform definition of the histological response to the treatment of EoE, and the histological criteria of the remission vary in different studies. Meanwhile, the development of uniform histological criterion of remission will allow for the results of treatment to be evaluated more objectively and to compare the results of different studies.

In some studies, the histological criterion of the remission is PEC < 15 eos/hpf [32,49,63,77,81,85]. Wolf W.A. et al. [85] showed that PEC < 15 was associated with endoscopic response in 90% of patients, with the decrease in symptoms in 88% of patients and with both clinical and endoscopic responses in 81% of patients. The logistic regression showed that a more prominent decrease in the eosinophil count is associated with a higher probability of the symptomatic response; for every 10% decrease in the eosinophil count, the symptom response increased by approximately 7% (*p* = 0.04), the endoscopic response increased by 6% (*p* < 0.001), and combined symptomatic and endoscopic responses increased by 10% (*p* = 0.01) [86]. PEC < 15 eos/hpf was a criterion for including studies in a meta-analysis [87]. This meta-analysis has shown that observational studies more often used the threshold of <15 eos/hpf, while interventional studies used more stringent criteria: <6 [32,49,54,70,71], <5 eos/hpf, and even <1 eos/hpf [53,88]. Gupta S.K., 2015 [71] defined a histological response as <6 eos/hpf, and histological remission as <1 eos/hpf.

Nevertheless, a low PEC is accompanied by the alleviation of symptoms only in a fraction of patients. In 85% of children with histological remission (PEC 0–5 eos/hpf), symptoms persisted [35]. Alexander J.A. et al. [36] revealed a histological response (>90% decrease in eosinophil count from baseline) in 61.9% of adults with EoE on fluticasone therapy and a symptomatic response in 42.9% of patients. Histological remission (<1 eos/hpf) was reached in 76.1% of patients in the group using high doses of oral budesonide suspension, but symptoms were absent only in 17.6% of patients [71]. Safroneeva E. et al. [37] showed that the area under the curve for symptoms measured by EEsAI and histological remission comprised 0.60 for PEC < 20/мм^2^ (which is equivalent to <5 eos/hpf) and 0.61 for PEC <60/мм^2^ (which is equivalent to <15 eos/hpf), and the accuracy rates were 62.1% and 61.7%, respectively. Therefore, PEC is not a sufficient histological criterion to define remission. Moreover, PEC reflects only inflammatory changes and does not reflect remodeling and stenofibrosis.

Aceves S.S. et al. [69] evaluated not only PEC, but other parameters in epithelium (basal zone hyperplasia (BZH), dilated intercellular spaces (DIS), desquamation of epithelium, clusters of eosinophils, and eosinophil degranulation) and in lamia propria (eosinophil count and fibrosis). The difficulty in using this score is that lamina propria does not necessary present in biopsy specimens. The rate of identification of the lamina propria in biopsies range from 38% to 75% in different studies [89,90,91]. But the advantage of this score is the evaluation of both—inflammatory features and the remodeling of the esophagus. The histological parameters of this score correlated with endoscopic features and dysphagia symptoms [69]. The epithelial remodeling score [92] is based on the previous score and includes BZH, DIS, and desquamation. The severity of the peak epithelial eosinophilia over the entire disease duration correlated positively with epithelial remodeling (r = 0.76, *p* < 0.0001), lamina propria eosinophilia (r = 0.54, *p* < 0.0001), and fibrosis (r = 0.53, *p* < 0.0001). The severity of lamina propria fibrosis (LPF) over time was lower in patients with PEC < 15 eos/hpf. Patients with a clinical response to topical corticosteroids demonstrated less prominent features of remodeling than non-responders.

Collins M.H. et al. [93] developed an eosinophilic esophagitis histological scoring system (EoEHSS), which includes the following parameters (Table 3): PEC, DIS, BZH, eosinophilic abscesses (EA), eosinophil surface layering (SL), surface epithelial alteration (SEA), dyskeratotic epithelial cells (DEC), and LPF (Figure 1). EoEHSS evaluates not only the severity of features (grade), but also the extension of histological changes (stage). It allows for EoE to be diagnosed and for the discrimination between active EoE and treatment status. Moreover, logistic regression shows better results of EoEHSS compared with PEC in identifying treatment status. Distinct parameters of EoEHSS show a correlation with clinical symptoms given the low PEC. Whelan K.A. et al. [94] demonstrated that BZH persisted almost in 30% patients with PEC < 15 eos/hpf and was associated with ongoing symptoms of EoE (odds ratio 2.14; 95% CI, 1.03–4.42; *p* = 0.041) and endoscopic features of EoE (odds ratio 7.10; 95% CI, 3.12–16.18; *p* < 0.001). It can be explained by the elevated count of mast cells in patients with BZH and DIS that induce an IgE-mediated allergic response and are the component of Th2 inflammation in EoE [95]. Bolton S.M. et al. [95] identified BZH in 48.4% of distal biopsies and in 45.1% of biopsies from the mid-esophagus. In this study, BZH was associated both with an elevated count of mast cells and with their degranulation. Interestingly, in patients with EoE without symptoms, the count of mast cells was lower than in patients with active EoE.

Even though lamina propria may not present in biopsies, the following combination of parameters can be an indirect indicator of the presence (but not stage) of LPF: age, BZH, DEC, and SEA [96]. The accuracy of this model for the identification of LPF comprised 84%. In other study, the strongest correlation was identified between LPF and two parameters of EoEHSS—DEC (r = 0.75; *p* < 0.001) and CEA (r = 0.70; *p* < 0.001) [91]. In logistic regression, the presence of DEC and CEA predicted LPF with an area under the curve of 0.91 (0.85–0.98) for CEA and 0.90 (0.83–0.97) for DEC.

Based on EoEHSS, Collins M.H. et al. [97] proposed the EoE histology remission score (EoEHRS) that includes two criteria: (1) PEC < 15 eos/hpf and (2) EoEHSS grade < 3 and EoEHSS stage < 3. The authors give the reason for their approach that PEC is not the only hallmark and component of inflammation in EoE and the evaluation of biopsies with EoEHSS may better define the remission. Moreover, the authors demonstrated a moderate correlation of EoEHRS with a decrease in symptoms of dysphagia (0.39–0.30, *p* ≤ 0.01), pain (0.48–0.34, *p* ≤ 0.01), and GERD symptoms (0.51–0.32, *p* ≤ 0.01) evaluated by PEESSv2.0. The correlation of PEC with symptom severity was weaker compared with EoEHRS. Moreover, biopsies that reached remission refined by EoEHRS showed a low expression of CPA3 (*p* = 0.0022) and thryptase (*p* = 0.0088), which confirms the low biological activity of EoE.

## 5. Multidisciplinary Approach to Estimate Severity of EoE

As it is shown in our review, there are different methods to evaluate clinical symptoms and different criteria of endoscopic and histological remission (Figure 2). Meanwhile, the data suggest that the evaluation of severity and the definition of the remission in EoE should be built on a combination of symptoms, endoscopic findings, and histology. Therefore, an instrument for the evaluation of EoE severity should be developed based on a multidisciplinary approach, using clinical, endoscopic, and histological features. A multidisciplinary international research group recently developed the Index of Severity of Eosinophilic Esophagitis (I-SEE) [98], grading the severity of EoE in points, where 0–6 correspond to mild EoE, 7–14 points correspond to moderate EoE, and ≥15 points correspond to severe EoE (Table 4). The I-SEE needs further validation and will be doubtlessly applied in clinical practice.

## 6. Conclusions

Currently, there is no uniform definition of the term “remission” in EoE. An assessment of the severity of EoE implies the evaluation of clinical, endoscopic, and histological features. EREFS is a gold standard for endoscopic examination in patients with EoE. EoEHSS is the most reliable histologic system for the evaluation of biopsies with EoE. The Multidisciplinary Severity Index for Eosinophilic Esophagitis (I-SEE) is promising but needs further validation.

## Figures and Tables

**Figure 1 biomedicines-11-03204-f001:**
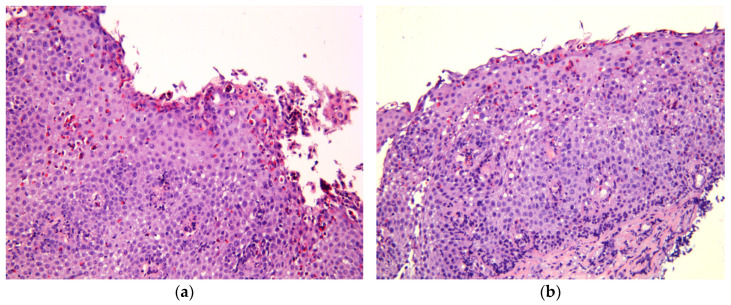
Histological features of EoE, hematoxylin, and eosin staining; magnification ×200. (**a**) Marked eosinophil surface layering (SL) with surface epithelial alteration (SEA), eosinophilic abscesses (EA). (**b**) Intraepithelial eosinophilic infiltration, pronounced basal zone hyperplasia (BZH), lamina propria fibrosis (LPF).

**Figure 2 biomedicines-11-03204-f002:**
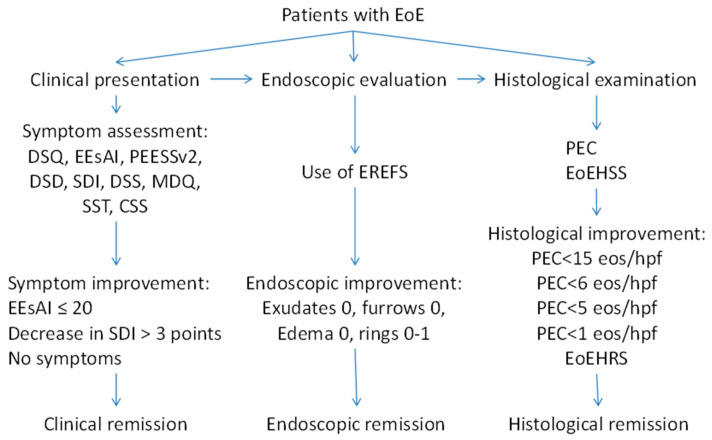
Different criteria of the remission in EoE.

**Table 1 biomedicines-11-03204-t001:** Comparison of different instruments for symptom assessment in EoE.

Features	DSQ	DSD	EEsAI	SDI	DSS	MDQ	PEESSv2	SST	CSS
Age									
For adults	+	+	+	+	+	+			
For children							+	+	+
Dysphagia									
Presence	+	+	+	+	+	+	+	+	+
Duration			+		+	+			
Frequency			+	+	+	+	+		
Severity	+	+	+	+	+	+	+		+
Presence of pain		+	+					+	
Behavior adaptation	+	+	+		+	+	+		
Heartburn						+	+	+	+
Regurgitation						+	+	+	+
Odynophagia						+		+	+
Presence of allergies or asthma						+			
Medication/treatment used						+			
Chest pain							+		
Abdominal pain							+	+	+
Vomiting							+	+	+
Nausea							+	+	+
Poor appetite							+		
Nocturnal awakening								+	+
Gastrointestinal hemorrhage								+	
Anorexia or early satiety								+	+

**Table 2 biomedicines-11-03204-t002:** EREFS score [73].

Endoscopic Feature	Description	Grade
Major features		
Fixed rings	None	0
	Mild (subtle circumferential ridges)	1
	Moderate (distinct rings that do not impair passage of a standard diagnostic adult endoscope (outer diameter 8–9.5 mm))	2
	Severe (distinct rings that do not permit passage of a diagnostic endoscope)	3
Exudates (white spots, plaques)	None	0
	Mild (lesions involving <10% of the esophageal surface area)	1
	Severe (lesions involving >10% of the esophageal surface area)	2
Furrows (vertical lines, longitudinal furrows)	Absent	0
	Present	1
Edema (vascular pattern, mucosal pallor)	Absent (distinct vascularity present)	0
	Loss of clarity or absence of vascular markings	1
Stricture	Absent	0
	Present	1
Minor features		
Crepe paper esophagus	Absent	0
	Present	1

**Table 3 biomedicines-11-03204-t003:** Eosinophilic esophagitis histological scoring system (EoEHSS) [93].

Histological Criteria	Grade Score	Stage Score
Eosinophilic inflammation	0—Intraepithelial eosinophils not present	0—Intraepithelial eosinophils 0–14/hpf,
	1—PEC < 15/hpf	1—PEC ≥ 15/hpf in <33% of hpfs
	2—PEC 15–59/hpf	2—PEC ≥ 15/hpf in 33–66% of hpfs
	3—PEC > 60/hpf	3—PEC ≥ 15/hpf in >66% of hpfs
Epithelial basal zone	0—BZH not present	0—BZH not present
	1—basal zone occupies > 15% but <33% of total epithelial thickness	1—BZH (any grade > 0) in <33% of epithelium
	2—basal zone occupies 33–66% of total epithelial thickness	2—BZH (any grade > 0) in 33–66% of epithelium
	3—basal zone occupies > 66% of total epithelial thickness	3—BZH (any grade > 0) in >66% of epithelium
Eosinophil abscess (EA)	0—groups or aggregates of eosinophils not present	0—groups or aggregates of eosinophils not present
	1—group of 4–9 eosinophils	1—EA (any grade > 0) in <33% of epithelium
	2—group of 10–20 eosinophils	2—EA (any grade > 0) in 33–66% of epithelium
	3—group of >20 eosinophils	3—EA (any grade > 0) in >66% of epithelium
Eosinophil surface layering (SL)	0—absent SL (fewer than 3 aligned eosinophils)	0—absent SL
	1—SL of 3–4 eosinophils	1—SL (any grade > 0) in <33% of epithelium
	2—SL of 5–10 eosinophils	2—SL (any grade > 0) in 33–66% of epithelium
	3—SL of >10 eosinophils	3—SL (any grade > 0) in >66% of epithelium.
Dilated intercellular spaces (DIS)	0—DIS not seen at any magnification	0—DIS not seen at any magnification
	1—Intercellular bridges in DIS visible at 400× magnification only	1—DIS (any grade > 0) in <33% of epithelium
	2—Intercellular bridges in DIS visible at 200× magnification	2—DIS (any grade > 0) in 33–66% of epithelium
	3—Intercellular bridges in DIS visible at 100× magnification or lower	3—DIS (any grade > 0) in >66% of epithelium
Surface epithelial alteration (SEA)	0—SEA not present	0—SEA not present
	1—SEA without eosinophils	1—SEA (any grade > 0) in <33% of epithelium
	2—SEA with any eosinophils	2—SEA (any grade > 0) in 33–66% of epithelium
	3—shed altered surface epithelium admixed with numerous eosinophils consistent with exudate	3—SEA (any grade > 0) in >66% of epithelium
Dyskeratotic epithelial cells (DECs)	0—DEC not present	0—DEC not present
	1—1 DEC/HPF	1—DEC (any grade > 0) in <33% of epithelium
	2—2–5 DEC/HPF	2—DEC (any grade > 0) in 33–66% of epithelium
	3—>5 DEC/HPF	3—DEC (any grade > 0) in >66% of epithelium
Lamina propria fibrosis (LPF)	0—LPF not present	0—LPF not present
	1—fibers are cohesive and interfiber spaces cannot be demarcated	1—LPF (any grade > 0) in <33% of lamina propria
	2—fiber diameter equals the diameter of a basal cell nucleus	2—LPF (any grade > 0) in 33–66% of lamina propria
	3—fiber diameter exceeds the diameter of a basal cell nucleus	3—LPF (any grade > 0) in >66% of lamina propria

**Table 4 biomedicines-11-03204-t004:** Eosinophilic Esophagitis Severity Index [97].

Points per Feature	1 Point	2 Points	4 Points	15 Points
Symptoms and complications				
Symptoms	Weekly	Daily	Multiple times per day or disrupting social functioning	
Complications	-	Food impaction with ER visit or endoscopy (patient ≥18 years)	Food impaction with ER visit or endoscopy (patient < 18 years) Hospitalization due to EoE	Esophageal perforation Malnutrition with body mass < 5th percentile or decreased growth trajectory Persistent inflammation requiring elemental formula, or systemic corticosteroid, or immunomodulatory treatments
Inflammatory features				
Endoscopy (edema, furrows, and/or exudates) Histology	Localized 15–60 eos/hpf	Diffuse >60 eos/hpf	- -	- -
Fibrostenotic features				
Endoscopy (rings, strictures)	Present, but endoscope passes easily	Present, but requires dilation or a snug fit when passing a standard endoscope	-	Cannot pass standard upper endoscope; repeated dilations (in an adult ≤ 18 years); or any dilation (in a child < 18 years)
Histology	-	BZH or LPF (or DEC/SEA if no LP)	-	-

## Data Availability

The data presented in this study are openly available in PubMed database.

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
