# Peer review of "Assessment of the Severity and the Remission Criteria in Eosinophilic Esophagitis"

_biomedicines, 2023, doi:10.3390/biomedicines11123204_

Round 1

Reviewer 1 Report

Comments and Suggestions for Authors

The esophagus is particularly suitable for an eosinophilic inflammatory process as an immune-related disease. On the contrary, a simple contract between gastrointestinal mucosa and an extraneous substance, like caffeine, can cause transitory eosinophilia in its superficial layer, probably activating the 'in loco' present eosinophils.  (seean experimental contribution: Nutrients, 2022; 14 (9): 1928. 

Reviewer 2 Report

Comments and Suggestions for Authors

Thanks to the authors for their effort in this review, although it still requires clarification of certain issues.

1. Introduction

The authors could include a brief paragraph explaining in more detail the types of treatments: proton pump inhibitors, corticosteroids, and food elimination diets.

2. Clinical symptoms assessment in EoE

- Other symptoms detected in patients with EoE should be included, such as nausea, weight loss, sleep problems.

- I recommend that the authors make a summary table with the main symptom severity indices and their main parameters.

3. Evaluation of endoscopic features in EoE

- Esophageal dilation and esophageal distensibility by functional luminal imaging probe (EndoFLIP) could be included as endoscopic methods.

Round 2

Reviewer 2 Report

Comments and Suggestions for Authors

Thanks to the authors for introducing the changes and the appropriate bibliography after my suggestions. The article would already be suitable for publication.